# Assessing the cost-effectiveness of economic strengthening and parenting support for preventing violence against adolescents in Mpumalanga Province, South Africa: An economic modelling study using non-randomised data

William E. Rudgard[1,2☯]*, Sopuruchukwu Obiesie[1☯], Chris Desmond[3], Marisa Casale[1,4], Lucie Cluver[1,5]

1 Department of Social Policy and Intervention, University of Oxford, Oxford, United Kingdom, 2 Centre for Social Science Research, University of Cape Town, Cape Town, South Africa, 3 School of Economics and Finance, University of Witwatersrand, Johannesburg, South Africa, 4 School of Public Health, University of the Western Cape, Cape Town, South Africa, 5 Department of Psychiatry and Mental Health, University of Cape Town, Cape Town, South Africa

☯ These authors contributed equally to this work.
* william.rudgard@spi.ox.ac.uk

## Abstract

There is limited evidence around the cost-effectiveness of interventions to reduce violence against children in low- and middle-income countries. We used a decision-analytic model to evaluate the cost-effectiveness of three intervention scenarios for reducing adolescent emotional, physical, and sexual abuse in Mpumalanga Province, South Africa. The intervention scenarios were: 1) Community grant outreach to link households to South Africa's Child Support Grant (CSG) if they are eligible, but not receiving it; 2) Group-based parenting support; and 3) Group-based parenting support 'plus' linkage to the CSG. We estimated average cost-effectiveness ratios (ACERs) for intervention scenarios over a ten-year time horizon, and compared them to a South Africa-specific willingness-to-pay (WTP) threshold (USD3390). Health effects were expressed in disability-adjusted life years (DALYs) averted. Our model considered four combinations of routine service versus trial-based costing, and population-average versus high prevalence of violence. Under routine service costing, ACERs for grant outreach and parenting support were below the WTP threshold when considering a population-average prevalence of violence USD2850 (Lower: USD1840-Upper: USD10,500) and USD2620 (USD1520-USD9800) per DALY averted, respectively; and a high prevalence of violence USD1320 (USD908-USD5180) and USD1340 (USD758-USD4910) per DALY averted, respectively. The incremental cost-effectiveness of parenting support plus grant linkage relative to parenting support alone was USD462 (USD346-USD1610) and USD225 (USD150-USD811) per DALY averted at a population-average and high prevalence of violence, respectively. Under trial-based costing, only the ACER for grant outreach was below the WTP threshold when considering a high prevalence of

**Data Availability Statement:** All relevant data are within the paper and its supporting information files.

**Funding:** Research reported in this publication was supported by the UK Research and Innovation Global Challenges Research Fund (UKRI GCRF) Accelerate Hub [ES/S008101/1 to LC]; the Oak Foundation [OFIL-20-057 to LC]; Wellspring Philanthropic Fund [Grant No. 16204 to LC]; and Oak Foundation/GCRF "Accelerating Violence Prevention in Africa" [R46194/AA001 to LC]. The funders had no role in study design, data collection and analysis, decision to publish, or preparation of the manuscript. For the purpose of Open Access, the author has applied a CC BY public copyright licence to any Author Accepted Manuscript version arising from this submission.

**Competing interests:** The authors have declared that no competing interests exist.

violence USD2580 (USD1640-USD9370) per DALY averted. Confidence intervals for all ACERs crossed the WTP threshold. In conclusion, grant outreach and parenting support are likely to be cost-effective intervention scenarios for reducing violence against adolescents if they apply routine service costing and reach high risk groups. Combining parenting support with grant linkage is likely to be more cost-effective than parenting support alone.

## Introduction

Annually, 1.4 billion children are estimated to experience violence or neglect worldwide, leading to more than 5.1 million disability-adjusted life years (DALYs) [1, 2]. The World Health Organization (WHO) and eight other global agencies have endorsed seven evidence-based strategies to address violence against children, known collectively as INSPIRE [3]. These seven strategies include **I**mplementation and enforcement of laws, **N**orms and values, **S**afe environments, **P**arent and caregiver support, **I**ncome and economic strengthening, **R**esponse and support services, and **E**ducation and life skills [3]. Further evidence is needed to help stakeholders prioritise strategies that are likely to have the greatest return on investment [4].

A previous study to guide the choice of 'best-buy' INSPIRE strategies in South Africa found that across seven INSPIRE-aligned protective factors, food security, caregiver supervision, and positive caregiving were the most promising targets for addressing multiple forms of violence against adolescents simultaneously [5]. Consistent with positive youth development theories, these three protective factors also combined additively, such that experiencing two or three of them together was associated with a significantly lower probability of experiencing multiple forms of violence, compared to experiencing one of them alone [5]. Research in Tanzania also suggests that parenting support and economic strengthening are likely to combine synergistically to reduce violence against children more than each intervention alone [6].

Evidence on the cost-effectiveness of interventions for reducing violence against children is essential for maximising the health gains from available resources [7]. However, economic evaluations of violence prevention interventions remain uncommon, particularly in low- and middle-income countries [8–11]. Experimental studies are recommended for evaluating cost-effectiveness, but they are expensive and complex, especially when more than one intervention is under consideration [12–14]. Decision analytic modelling is a cheaper alternative approach that uses secondary data sources, including published trials and observational studies, to model the expected costs and consequences of decision options [9, 12].

In this study, we aimed to use secondary data to estimate the cost-effectiveness of a selected list of evidence-based interventions to reduce violence against adolescents [15, 16]. We had three objectives: 1) Select expert- and evidence-based interventions for reducing violence against adolescents via improving food security, caregiver supervision, or positive caregiving in South Africa; 2) Gather together cost and effectiveness data relating to selected interventions from a variety of high-quality secondary data sources; 3) Combine these data in a decision-analytic model to estimate the cost-effectiveness of selected interventions.

## Methods

This economic modelling study builds on a non-randomised analysis of seven INSPIRE-aligned protective factors among adolescents in South Africa, which found that food security, caregiver supervision, and positive caregiving were associated with lower odds of multiple forms of adolescent violence victimisation including, emotional abuse, physical abuse, sexual

abuse, and community violence victimisation [5]. Our methodological approach involved three steps. First, we consulted with regional experts on preventing violence against adolescents to identify the best candidate interventions for reducing violence by improving food security, caregiver supervision, and positive caregiving in South Africa. Second, we drew together cost and effectiveness data for the selected interventions from published literature, online survey data, and further expert consultation. Third, we combined these data into a decision-analytic model to estimate the cost per DALY averted of our selected interventions. This choice enabled us to generate a single estimate of cost-effectiveness for selected interventions, but also limited us to only considering adolescent violence victimisation outcomes with evidence for their attributable DALYs in the published literature. Of the four kinds of adolescent violence victimisation investigated in the non-randomised analysis in South Africa, we found data on the DALYs attributable to emotional, physical, and sexual abuse, but not community violence victimization [5]. We include full details of data sources and assumptions throughout the analysis and report our study using the Consolidated Health Economic Evaluation Reporting Standard (CHEERS) checklist, S1 Checklist [17, 18].

## Study setting

The study setting was Mpumalanga Province, South Africa where intervention costs and adolescents' experience of violence are expected to lie between the other two provinces where data were collected for the non-randomised analysis of seven INSPIRE-aligned protective factors in South Africa; the poorer Eastern Cape Province, and richer Western Cape Province [5, 19].

## Study population

The study population was households with a monthly income per capita below the South African upper poverty line and an adolescent aged between 10 and 19 years [20–22]. In South Africa, the upper poverty line is approximately equal to the means-test threshold used to assess applicant's eligibility for the national Child Support Grant (CSG).

## Hypothesised scenarios for violence prevention

We consulted experts on violence prevention to identify the best candidate interventions for reducing adolescent violence via improving food security, caregiver supervision, and positive caregiving in South Africa. Our focus on these protective factors was based on evidence of their association with a lower probability of multiple forms of violence against adolescents in South Africa [5]. Criteria for candidate interventions were that (i) there should be existing or early implementation within South Africa; and ii) evidence supporting their impact on either food security or positive and supervisory caregiving in Southern Africa. Key details on intervention implementation, staffing, and duration are provided in S1 Table [5].

**Scenario 1: Community outreach programme to link households that are eligible but not receiving South Africa's CSG [Grant outreach].** It is estimated that around 18% of children eligible for anti-poverty social grants in South Africa do not access this support [23, 24]. Regional evidence supports the effectiveness of cash grants in addressing food security [25, 26], and evidence from South Africa supports the potential of community initiatives to link eligible households to social grants [27]. Therefore, in our analysis, we modelled the impact of a community outreach programme implemented over 17-months during which paraprofessional social workers would actively liaise with community leaders and networks to screen for households that may be eligible for the CSG but don't already receive it. Eligible households not already receiving the CSG would then be supported to access it.

**Scenario 2: Parenting support programme based on WHO/UNICEF's Parenting for Lifelong Health (PLH) programme [Parenting support].**   Across South Africa, parenting support programmes are offered by a variety of non-profit organisations, including in Mpumalanga Province, South Africa as part of the Mothers2Mothers (M2M) Children and Adolescents are My Priority (CHAMP) project [28, 29]. Evidence also supports the potential of parenting support programmes to improve caregiver supervision in South Africa [30]. In our analysis, we thus modelled the roll-out of a parenting support programme akin to the Parenting for Lifelong Health Teen (PLH Teen), which was co-founded with WHO and UNICEF, and is structured as a 14-session intervention for small family groups. Participants are recruited into the programme by self or community-referral using the two screening questions: 'do you and your teen argue and shout a lot?' and 'do you sometimes end up hitting your teen when things are really stressful?' [30, 31].

**Scenario 3: Integrated parenting support programme plus component to link households that are eligible but not receiving South Africa's CSG [Parenting support plus grant linkage].**   In this combined scenario, we sought to explore the additive benefits of an intervention that acts on two protective factors. We modelled the roll-out of a parenting support plus grant linkage programme, structured similarly to the 14-session PLH Teen programme but including one additional session during which facilitators would assess families for their eligibility for inclusion in South Africa's CSG. Families found to be eligible but not receiving the CSG would be linked to the relevant social services to support them in accessing the grant. This parenting support plus grant linkage intervention would be distinct from the community grant outreach intervention, as it would focus on families that attend the parenting support intervention rather than the broader community.

## Choice of model

To estimate the cost-effectiveness of selected scenarios, we constructed a probability tree model that modelled i) the effect of selected interventions on food security and/or caregiver supervision, and ii) the corresponding effect of the estimated improvement in these intermediary protective factors on adolescent violence victimisation, Fig 1. This type of model was a simple and logical way for modelling the effect of selected interventions on violence outcomes via intermediary protective factors. We did not model parenting support or parenting support 'plus' grant linkage acting via positive caregiving as there is mixed evidence for whether group-based parenting support programmes improve this intermediary protective factor. Also, we did not model parenting support as reducing sexual abuse as there was no evidence that higher caregiver supervision was associated with this violence outcome in the observational analysis of seven INSPIRE-aligned protective factors that informed this study [5]. With these considerations, we modelled grant outreach acting on emotional, physical, and sexual abuse via food security, parenting support acting on emotional and physical abuse via caregiver supervision, and parenting support plus grant linkage acting on emotional, physical, and sexual abuse via both food security and caregiver supervision.

## Data sources

Model inputs were drawn from published literature, online survey data, and expert consultation, Table 1, S2 Table. Estimates of the percentage of adolescents in Mpumalanga living below the upper poverty line were derived from South Africa's 2018 General Household Survey [32]. Estimates of the prevalence of emotional, physical, and sexual abuse among adolescents were obtained from the non-randomised analysis of seven INSPIRE-aligned protective factors for adolescent violence in South Africa [5].

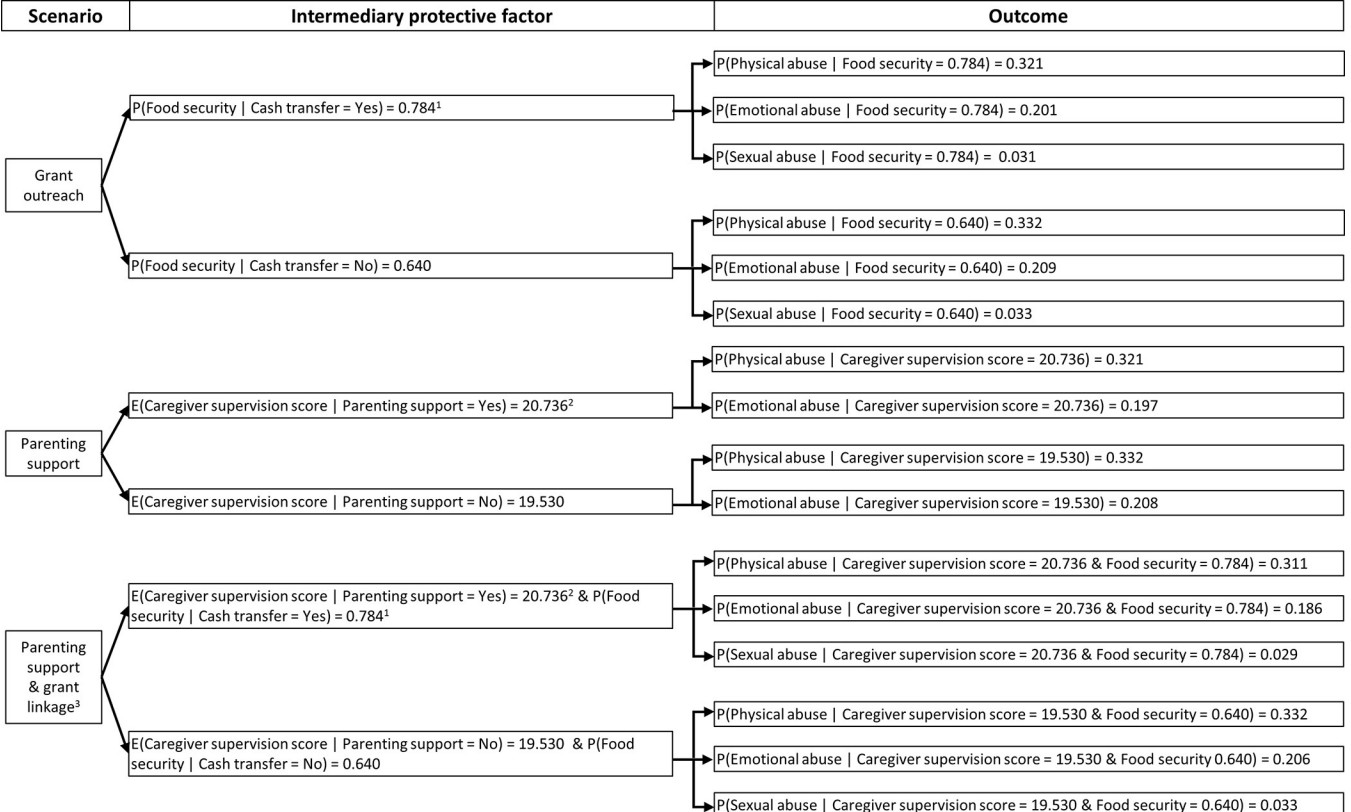

**Fig 1. Probability tree model for estimating the effect of intervention scenarios on adolescent violence outcomes via intermediary protective factors.**
[1]Estimate synthesized across two studies examining the impact of grants on household food security in sub-Saharan Africa and crosschecked using data from South Africa's 2018 General Household Survey. [2]Estimate obtained from the primary analysis of a randomised evaluation of PLH Teen in South Africa. [3]This tree applies only to a subset of adolescents from households in the parenting plus grant linkage scenario who, having been identified as eligible and not currently receiving the CSG. These adolescents would benefit from enhanced caregiver supervision and food security. The remaining adolescents would only benefit from enhanced caregiver supervision as in scenario two.

**Intervention effects.** Estimates of intervention effectiveness were drawn from the published literature. For community grant outreach, we generated a pooled estimate for the effect of cash transfers on food security by meta-analysis of two studies identified using a rigorous literature review, S1 Text [25, 26]. For parenting support, we used estimates of the effect of the PLH Teen intervention on caregiver supervision taken from the original randomised evaluation in South Africa [30]. For parenting support plus grant linkage, we used our pooled estimate for the effect of cash transfers on food security and our estimate of the effect of PLH Teen on caregiver supervision. Finally, for the effect of enhanced levels of food security and caregiver supervision on adolescent violence victimisation, we used estimates from the non-randomised analysis of seven INSPIRE-aligned protective factors for reducing adolescent violence in South Africa [5]. We modelled food security and caregiver monitoring as combining additively to reduce adolescent violence victimisation, since there was no evidence in the original analysis that they combine multiplicatively with respect to violence against adolescents [5].

**Intervention costs.** All three scenarios were costed from the provider's perspective using the ingredients method. Unit costs and quantities for parenting support and parenting support plus grant linkage interventions were based on data from the PLH Teen trial in South Africa and PLH Teen implementation in Thailand and Tanzania [6, 30, 33]. We were unable to find cost data on an intervention similar to the modelled grant outreach intervention, so unit costs

**Table 1. Summary of key model inputs, their description, and source.**

| | Value | Description | Sources |
|---|---|---|---|
| **Mpumalanga Province** | | | |
| Total population | 4,523,000 | | Mid-year population estimate 2018, Stats SA |
| Adolescent population | 18% | | Mid-year population estimate 2018, Stats SA |
| Adolescents living below the UPL | 35% | UPL = Monthly household income per capita < USD 79 | South Africa GHS 2018, Stats SA |
| Number of adolescents per household | 1.03 | | South Africa GHS 2018, Stats SA |
| Number of children per household | 2.56 | | South Africa GHS 2018, Stats SA |
| **Intervention scenario 1: Grant outreach** | | | |
| Eligible children receiving a CSG | 1,105,791 | Children eligible for the CSG are <18 years | SASSA, 2019 |
| Eligible children not receiving CSG | 18% | | UNICEF, 2016; South Africa GHS 2018, Stats SA |
| Number of household visits needed to identify eligible adolescent | 3 | | Expert consultation |
| Success rate in linking eligible adolescents to CSG | 70% | | Expert consultation; Thurman et al. 2015 |
| Number of household visits, per paraprofessional social worker, per day | 3 | Travel by public transport/ walking | Expert consultation |
| Intervention duration | 17 months | Implementation is ongoing until all eligible adolescents not receiving CSG have been visited | Derived based on intervention staffing and number of household visits needed to identify eligible adolescent not receiving CSG |
| **Staff** | | | |
| Paraprofessional social workers | 1000 | Human resources needed to link 6% of eligible households per month | Staffing needs were based on a target of linking 6% of eligible adolescents to the CSG per month, and a hierarchical organisational structure similar to PLH Teens. Data from the PLH Teen trial were obtained through personal communication with Dr Jamie Lachmann |
| Supervisors | 50 | 1 per 20 paraprofessional social workers | |
| District coordinators | 3 | 1 per district | |
| Provincial representatives | 1 | 1 per province | |
| **Effectiveness** | | | |
| Effect on food security[1] | 1.22 (1.06; 1.34) | RR (95% CI) | Meta-analysis of cash transfer effects on household food security Handa et al. 2022 and Bhalla et al. 2018 |
| Effect on adolescent violence | | | |
| Emotional abuse | 0.96 (0.93; 0.99) | RR (95% CI) | Observational analysis of protective factors and adolescent violence by Cluver & Rudgard et al. 2020 |
| Physical abuse | 0.97 (0.95; 0.99) | | |
| Sexual abuse | 0.94 (0.89; 0.98) | | |
| **Cost, USD** | | | |
| Staff training, per staff | 38; 62 | Routine service; Trial-based estimate | Unit costs were based on data from the PLH Teen trial obtained through personal communication with Dr Jamie Lachmann, and unit quantities were based on expert consultation |
| Community outreach, per household visit | 62; 113 | | |
| **Intervention scenario 2: Parenting support** | | | |
| Eligible families targeted, per year | 5% | Based on PLH Teen scale-up in Thailand and Tanzania | PLH Teen data obtained through personal communication with Dr Jamie Lachmann |
| Family uptake success rate | 90% | We expect 5% lower compliance in a real-world setting than in a trial | Randomised evaluation of PLH Teen by Cluver et al. 2018 |
| Number of families, per family group | 15; 25 | Routine service; Trial-based estimate | PLH Teen data obtained through personal communication with Dr Jamie Lachmann |
| Duration of implementation | 10 years | Implementation is continuous over the study time horizon | |
| **Staff** | | | |

*(Continued)*

**Table 1.** (Continued)

|  | Value | Description | Sources |
|---|---|---|---|
| Facilitators | 91; 151 | Routine service; Trial-based estimate | Staffing needs were based on the hierarchical organisational structure used by PLH Teen. Data on PLH Teen staffing were obtained through personal communication with Dr Jamie Lachmann |
| Coaches | 9; 15 | 6; 6 family groups per facilitator per year | |
| Coordinators | 5; 8 | 10; 10 facilitators per coach | |
| Assistant coordinators | 9; 30 | 20; 20 facilitators per coordinator | |
| District coordinators | 3; 3 | 5; 10 facilitators per assistant coordinator | |
| Provincial representatives | 1; 1 | 1 district per district coordinator 1 province per provincial representative | |
| **Effectiveness** | | | |
| Effect on caregiver supervision | 1.21 (2.09; 0.32) | Effect size (95% CI) | Randomised evaluation of PLH Teen by Cluver et al. 2018 |
| Effect on adolescent violence | | | |
| Emotional abuse | 0.95 (0.91; 0.99) | RR (95% CI) | Observational analysis of protective factors and adolescent violence by Cluver & Rudgard et al. 2020 |
| Physical abuse | 0.97 (0.94; 0.99) | | |
| Sexual abuse[1] | - | | |
| **Cost, USD** | | | |
| Staff training, per staff | 112; 365 | Routine service; Trial-based estimate | PLH Teen data obtained through personal communication with Dr Jamie Lachmann |
| Delivery cost, per family group | 33; 201 | | |
| Office space and equipment, per year | 6300; 38,000 | | |
| **Intervention scenario 3: Parenting support plus grant linkage** | | | |
| Eligible families targeted, per year | 5% | Based on PLH Teen scale-up in Philippines and Thailand | PLH Teen data obtained through personal communication with Dr Jamie Lachmann |
| Family uptake success rate | 90% | We expect 5% lower compliance in a real-world setting than in a trial | Randomised evaluation of PLH Teen by Cluver et al. 2018 |
| Number of families, per family group | 15; 25 | Routine service; Trial-based estimate | PLH Teen data obtained through personal communication with Dr Jamie Lachmann |
| Duration of implementation | 10 years | Implementation is continuous over the study time horizon | |
| Eligible children not receiving CSG | 18% | | UNICEF, 2016; South Africa GHS 2018, Stats SA |
| Success rate in linking eligible adolescents to CSG | 70% | | Expert consultation; Thurman et al. 2015 |
| **Effectiveness** | | | |
| Effect on caregiver supervision | 1.21 (0.32; 2.09) | Effect size (95% CI) | Randomised evaluation of PLH Teen by Cluver et al. 2018 |
| Effect on food security | 1.22 (1.06; 1.34) | RR (95% CI) | Meta-analysis of cash transfer effects on household food security Handa et al. 2022 and Bhalla et al. 2018 |
| Effect on adolescent violence | | | |
| Emotional abuse | 0.95 (0.84; 0.97) | RR (95% CI) | Observational analysis of protective factors and adolescent violence by Cluver & Rudgard et al. 2020 |
| Physical abuse | 0.94(0.89; 0.98) | | |
| Sexual abuse | 0.89 (0.82; 0.97) | | |
| **Staff** | | | |
| Facilitators | 91; 151 | Routine service; Trial-based estimate | Staffing needs were based on the hierarchical organisational structure used by PLH Teen. Data on PLH Teen staffing were obtained through personal communication with Dr Jamie Lachmann |
| Coaches | 9; 15 | 6; 6 family groups per facilitator per year | |
| Coordinators | 5; 8 | 10; 10 facilitators per coach | |
| Assistant coordinators | 9; 30 | 20; 20 facilitators per coordinator | |
| District coordinators | 3; 3 | 5; 10 facilitators per assistant coordinator | |
| Provincial representatives | 1; 1 | 1 district per district coordinator 1 province per provincial representative | |

(Continued)

**Table 1.** (Continued)

| | Value | Description | Sources |
|---|---|---|---|
| **Cost, USD** | | | |
| Staff training, per staff | 190; 370 | Routine service; Trial-based estimate | PLH Teen data obtained through personal communication with Dr Jamie Lachmann |
| Delivery cost, per family group | 35; 208 | | |
| Office space and equipment, per year | 6600; 39,400 | | |

[1]Violence outcome was not considered for the parenting support scenario, as there was no evidence that it was significantly associated with caregiver supervision.
Abbreviations: UPL, upper poverty line; Stats SA, Statistics South Africa; CSG, Child Support Grant; RR, relative risk; CI, confidence interval; GHS, General Household Survey; USD, United States dollar; PLH, Parenting for Lifelong Health

were based on the PLH Teen trial and unit quantities based on expert consultation. We did not consider the operational costs of delivering the CSG in our analysis (i.e. the value of grants, overheads) as it was agreed with experts that these costs are already budgeted for by the South African government in its annual projection for the expected number of households that are eligible to receive the grant.

**Data management.** All costs captured before 2021 were adjusted for inflation and converted to USD using the conversion rate of USD 1 = South African Rand 14.93.

## Data analysis

We modelled the costs and DALYs due to violence victimisation averted for the three intervention scenarios over a time horizon of ten years, as recommended by WHO-CHOICE guidelines.

**Modelling the effectiveness of interventions.** For each intervention scenario, first, we used our probability tree model to estimate the relative risk of emotional, physical, and sexual abuse comparing the probability of these outcomes in the presence and absence of cash transfers and/ or parenting support, Table 1. We used the 95% confidence intervals around relative risk ratios to estimate upper and lower bounds around our main effects. Second, we estimated the cumulative number of cases averted due to intervention activities, accounting for an annual dropout rate from grant outreach (2%), an attrition rate from parenting support (3%), an annual ageing out effect for grant outreach (13%) and parenting support (11%), and an annual decay in the intervention's effects on food security (3%) and caregiver monitoring (33%) [34–37]. Effects of receiving the CSG were modelled to cease when adolescents turned 18 and became ineligible to receive this service. Third, we converted estimated subtotals of cases of violence averted into DALYS averted and summed across them to estimate the total DALYS averted for each intervention. DALYs attributable to emotional, physical, and sexual abuse were calculated from the only known evaluation of the economic consequences of violence against children in South Africa, and estimates of the prevalence of violence against adolescents in South Africa, S3 Table [11, 38, 39]. For full details on this method, see Redfern et al., 2019 [11].

**Modelling the cost of interventions.** For each intervention scenario, we estimated the total cost of modelled intervention activities over ten years. For grant outreach, this was the cost of a one-off 17-month community outreach initiative to link households that are eligible to receive South Africa's CSG but do not currently receive it. The grant outreach programme would end once all eligible but excluded households had been visited. For parenting support and parenting support plus grant linkage, this was the total cost of sequential rounds of 14-week parenting support programmes over ten years without and with grant linkage, respectively. For all three scenarios, we applied an annual discount rate of 4% to costs, as recommended for economic evaluations in upper middle-income countries [40]. We also estimated

the cost of interventions using 'routine service' and 'trial-based' ingredient costing. These two costings targeted the same number of families but varied in staff salaries, with the latter providing significantly higher salaries, and budgeting for venue hire, food for participants, printed training materials, and communication with participants via mobile phones. Routine service costing was based on daily salary rates equivalent to South Africa's average monthly earnings in March 2021, while trial-based costing was based on daily salary rates three times higher than this [41].

**Modelling the cost-effectiveness of interventions.** We calculated average cost-effectiveness ratios (ACERs) for intervention scenarios by dividing intervention total costs by total DALYs averted. We calculated incremental cost-effectiveness ratios (ICERs) for parenting support plus grant linkage relative to standalone parenting support by dividing the two scenarios difference in cost by the difference in DALYs due to violence victimisation averted [12, 18]. Estimated cost-effectiveness ratios and their upper and lower bounds were evaluated against a South African-specific willingness-to-pay (WTP) threshold estimated by Edoka and Stacey, equivalent to USD 3390 [42].

We calculated cost-effectiveness ratios for all three of our interventions under routine service and trial-based costings, and at two estimates of the prevalence of adolescent violence outcomes. These were 'population-average' prevalence, which was equal to the rates observed in our source reference [5]; and 'high' prevalence, which was equal to double the rates observed in our source reference [5]. The rationale for modelling a high prevalence of violence was that adolescents targeted by our grant-outreach intervention (i.e., eligible but not receiving the CSG), and our parenting support and parenting support plus grant linkage interventions (i.e., screened for regular arguments and/or physical abuse against children) are likely to experience higher than average levels of vulnerability and adolescent violence victimisation.

**Robustness checks.** First, we evaluated the robustness of our findings to using South Africa's GDP per capita for 2021 as our WTP threshold [18, 43]. Second, in the absence of a published evaluation of the effect of South Africa's CSG on food security, we also checked the robustness of our findings for the cost-effectiveness of community grant outreach, and parenting support plus grant linkage to a country-specific estimate for the relationship between income and food security estimated using South Africa's 2018 General Household Survey [44]. Further details of this secondary analysis are included in S2 Text.

## Ethics

This study used secondary data published in the public domain throughout the analysis.

## Results

Population estimates for Mpumalanga Province and the expected number of beneficiaries for the three intervention scenarios are summarized in Table 2. We estimated that approximately 434,000 adolescents are living below the poverty line in Mpumalanga. Under the grant outreach intervention scenario, 57,100 adolescents living in 55,500 households would be linked to the CSG. Under the parenting support intervention scenario, 66,900 adolescents living in 65,300 households would attend parenting support. Under the parenting support plus grant linkage scenario, 66,900 adolescents living in 65,300 households would attend parenting support, and 8200 of these would also be linked to the CSG.

### Estimated intervention effects

Estimates for the number of averted cases of violence victimisation and averted DALYS from the grant outreach, parenting support, and parenting support plus grant linkage scenarios are

**Table 2. Summary of provincial population estimates and the expected number of beneficiaries for each intervention scenario based on model inputs.**

|  | Estimate |
|---|---|
| **Mpumalanga Province** | |
| Adolescents | 667,000 |
| Adolescents living below the UPL | 434,000 |
| Adolescents receiving the CSG | 374,000 |
| **Intervention scenario 1: Grant outreach** | |
| Eligible adolescents not receiving the CSG | 77,000 |
| Eligible adolescents linked to CSG | 57,100 |
| Eligible households linked to CSG | 55,500 |
| **Intervention scenario 2: Parenting support** | |
| Adolescents attending parenting support | 66,900 |
| Households attending parenting support | 65,300 |
| **Intervention scenario 3: Parenting plus grant linkage** | |
| Adolescents attending parenting support | 66,900 |
| Households attending parenting support | 65,300 |
| Eligible adolescents not receiving the CSG | 11,700 |
| Eligible adolescents linked to CSG | 8200 |

All estimates are rounded to three significant figures.

Abbreviations: UPL, upper poverty line; CSG, Child Support Grant.

reported in Table 3. Our model estimated that through improvements in food security, the grant outreach scenario would avert 1180 (Lower: 320; Upper: 1830) and 2360 (649; 3700) DALYs attributable to emotional, physical, and sexual abuse, at population-average and high prevalence of violence, respectively. Through improvements in caregiver monitoring, the parenting support scenario would avert 995 (254; 1640) and 1910 (509; 3300) DALYs attributable to emotional and physical abuse, at population-average and high prevalence of violence, respectively. Through improvements in caregiver supervision and food security, the parenting support and grant linkage would avert 1150 (310; 1900) and 2310 (620; 3900) DALYs attributable to emotional, physical, and sexual abuse, at population-average and high prevalence of violence, respectively.

## Estimated intervention costs

Estimates for the costs associated with intervention scenarios are reported in Table 3. A more detailed breakdown of costs is also provided in S4 Table. Grant outreach was estimated to cost USD 59 and USD 106 per adolescent, and USD 3,360,000 and USD 6,080,000 at the provincial level, under routine service and trial-based costing, respectively. Parenting support was estimated to cost USD 36 and USD 215 per adolescent beneficiary, and USD 2,500,000 and USD 14,900,000 at the provincial level, under routine service and trial-based costing, respectively. The major cost drivers were staff salaries (80% and 54% of total costs for routine service and trial-based, respectively), and food for participants and facilitators (0% and 21% of total costs under the routine service and trial-based, respectively). Parenting support plus grant linkage was estimated to cost USD 38 and USD 223 per adolescent, and USD 2,590,000 and USD 15,400,000 at the provincial level, under routine service and trial-based costing, respectively.

## Intervention cost-effectiveness

ACERs and ICERS for intervention scenarios are summarised in Table 3. ACERS are also summarised against a South African-specific WTP threshold in Fig 2. An intervention is

**Table 3. Summary of the estimated cases of violence victimisation, DALYs averted, costs, and cost-effectiveness ratios of intervention scenarios over a ten-year time horizon.**

| | Averted cases of violence[1] (Lower; Upper) | DALYs averted (Lower; Upper) | Total cost per adolescent, USD | Total cost provincial, USD | ACER (Lower; Upper) | ICER (Lower; Upper) |
|---|---|---|---|---|---|---|
| **Routine service costing** | | | | | | |
| **Population-average prevalence of violence** | | | | | | |
| Grant outreach | 4540 (1250; 6970) | 1180 (320; 1830) | 59 | 3,360,000 | 2850 (1840; 10,500) | 2850 (1840; 10,500) |
| Parenting support | 3700 (987; 6350) | 955 (254; 1640) | 36 | 2,500,000 | 2620 (1520; 9800) | 2620 (1520; 9800) |
| Parenting support plus grant linkage | 4470 (1200; 7500) | 1150 (310; 1900) | 38 | 2,590,000 | 2250 (1363; 8350) | 462 (346; 1610) |
| **High prevalence of violence** | | | | | | |
| Grant outreach | 9100 (2500; 13,920) | 2360 (649; 3700) | 59 | 3,360,000 | 1420 (908; 5180) | 1420 (908; 5180) |
| Parenting support | 7410 (1970; 12,700) | 1910 (509; 3300) | 36 | 2,500,000 | 1320 (758; 4910) | 1320 (758; 4910) |
| Parenting support plus grant linkage | 8900 (2400; 15,000) | 2310 (620; 3900) | 38 | 2,590,000 | 1120 (664; 4180) | 225 (150; 811) |
| **Trial-based costing** | | | | | | |
| **Population-average prevalence of violence** | | | | | | |
| Grant outreach | 4540 (1250; 6970) | 1180 (320; 1830) | 106 | 6,080,000 | 5150 (3320; 19,000) | 5150 (3320; 19,000) |
| Parenting support | 3700 (987; 6350) | 955 (254; 1640) | 215 | 14,900,000 | 15,600 (9090; 58,700) | 15,600 (9090; 58,700) |
| Parenting support plus grant linkage | 4470 (1200; 7500) | 1150 (310; 1900) | 223 | 15,400,000 | 13,400 (8110; 49,700) | - |
| **High prevalence of violence** | | | | | | |
| Grant outreach | 9100 (2500; 13,920) | 2360 (649; 3700) | 106 | 6,080,000 | 2580 (1640; 9370) | 2580 (1640; 9370) |
| Parenting support | 7410 (1970; 12,700) | 1910 (509; 3300) | 215 | 14,900,000 | 7800 (4520; 29,300) | 7800 (4520; 29,300) |
| Parenting support plus grant linkage | 8900 (2400; 15,000) | 2310 (620; 3900) | 223 | 15,400,000 | 6670 (3950; 24,800) | - |

All estimates are rounded to three significant figures. We did not estimate ICERs for parenting support plus grant linkage when the ACER for parenting support alone was above our WTP threshold of USD3390.

[1]Forms of violence for grant outreach and parenting support plus grant outreach include emotional, physical, and sexual abuse, and for parenting support include emotional and physical abuse.

Abbreviations: DALY, disability-adjusted life years; USD, United States dollar; ACER, average cost-effectiveness ratio; ICER, incremental cost-effectiveness ratio.

considered cost-effective if it's ACER lies below the WTP threshold of USD 3390. The further an ACER lies to the right of the plane while remaining below the WTP threshold the higher its indicated cost-effectiveness.

**Under routine service costing.** At a population-average prevalence of violence, ACERs for grant outreach, parenting support, and parenting support plus grant linkage scenarios were all below the WTP threshold at USD 2850 (1840; 10,500), USD 2620 (1520; 9800), and USD 2250 (1363; 8350) per DALY averted, respectively. For all three scenarios, upper estimates of ACERs crossed the WTP threshold. The parenting support plus grant linkage scenario had an ICER of 225 (150; 811) per DALY averted compared to the parenting support alone scenario.

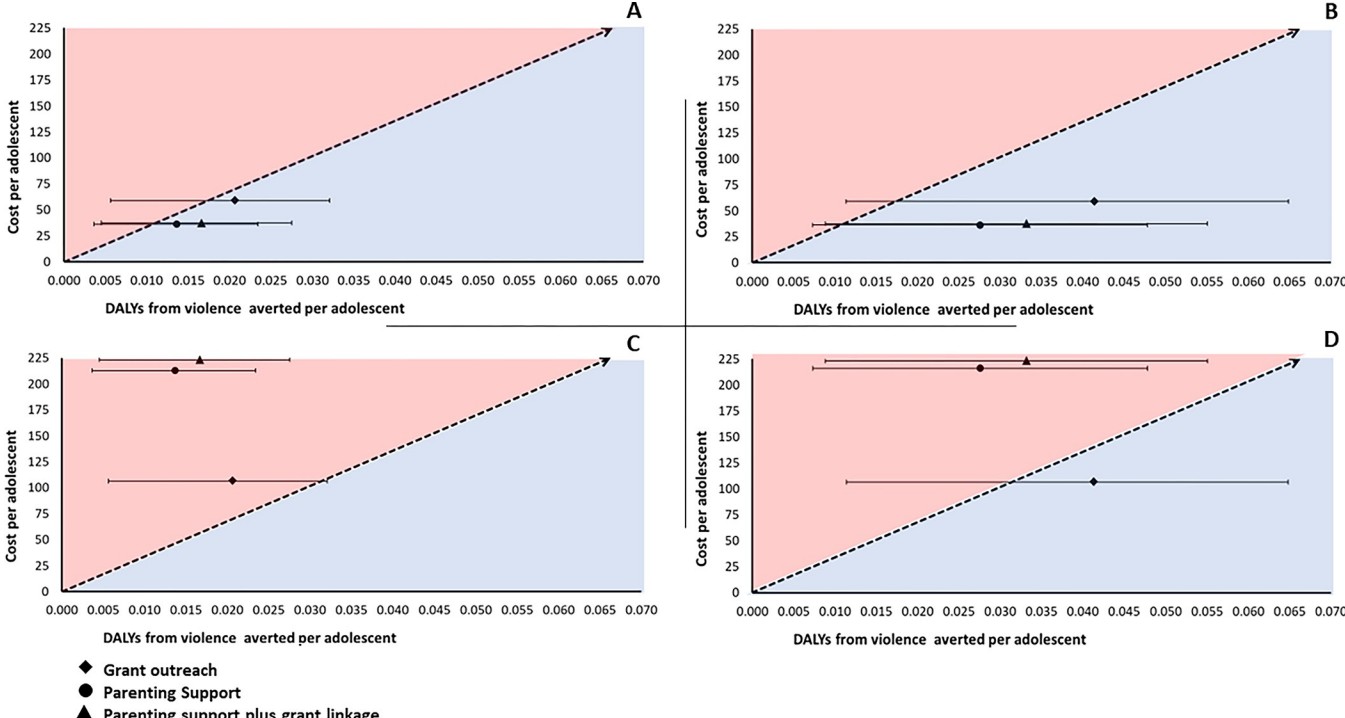

**Fig 2.** Cost-effectiveness plane scatter plots of costs per adolescent over effectiveness (in DALYs averted) for four scenarios at A) Routine service costing and population-average prevalence of violence, B) Routine service costing and high prevalence of violence, C) Trial-based costing and population-average prevalence of violence, and D) Trial-based costing and population-high prevalence of violence. Diagonal lines indicate an evidence-based SA-specific willingness to pay threshold for health interventions estimated by Edoka and Stacey, 2020. Strategies become increasingly cost-effective as they lie to the north easternmost part of the cost-effectiveness plane below the willingness to pay threshold. Strategies above the line are not cost-effective at the stated threshold. Abbreviations: DALYs, disability-adjusted life years; USD, United States dollar.

At a high prevalence of violence, ACERs for grant outreach, parenting support, and parenting support plus grant linkage scenarios were all below the WTP threshold at USD 1420 (908; 5180), USD 1320 (758; 4910), and USD 1120 (664; 4180) per DALY averted, respectively. For all three scenarios, upper estimates of ACERs crossed the WTP threshold. The parenting support plus grant linkage scenario had an ICER of USD 225 (150; 811) per DALY averted compared to the parenting support alone scenario.

**Under trial-based costing.**   At a population-average prevalence of violence, ACERs for grant outreach, parenting support, and parenting support plus grant linkage scenarios were all above the WTP threshold at USD 5150 (3320; 19,000), USD 15,600 (9090; 58,700), and USD 13,400 (8110; 49,700) per DALY averted, respectively. For the grant outreach scenario, the lower ACER estimate crossed the WTP threshold. Since the ACER for parenting support alone was above our WTP threshold, we did not estimate the ICER for parenting support plus grant linkage.

At a high prevalence of violence, the ACER for grant outreach scenario was below the WTP threshold at USD 2580 (1640; 9370) per DALY averted, but the ACERs for parenting support and parenting support plus grant linkage scenarios were above the WTP threshold at USD 7800 (4520; 29,300) and USD 6670 (3950; 24,800) per DALY averted. For the grant outreach scenario, the upper ACER estimate crossed the WTP threshold. Since the ACER for parenting support alone was above our WTP threshold, we did not estimate the ICER for parenting support plus grant linkage.

### Sensitivity analysis and robustness check

When using the WHO CHOICE WTP threshold based on South African GDP per capita instead of a South Africa-specific WTP threshold, results for routine service costing were the same except that at a high prevalence of violence, the lower estimates for ACERs did not cross the WTP threshold. Results for trial-based costing were also the same except that at a population-average prevalence of violence, the ACER for grant outreach was below the WTP threshold, S1 Fig. Substituting our pooled estimate for the effect of cash transfers on food security with an estimate of the relationship between income and food security from South Africa's 2018 General Household Survey, the findings for parenting support plus grant linkage remain consistent, but grant outreach ceases to be cost-effective, S2 Fig.

## Discussion

We find that decision-analytic modelling is a valuable and low-cost approach for assessing the cost-effectiveness of three expert- and evidence-informed interventions aimed at reducing adolescent violence victimisation in Mpumplanga, South Africa. Our model indicates that so long as routine service costing is used, investments in grant outreach, and parenting support may be cost-effective for reducing adolescent violence victimisation. Wide confidence intervals around our effect estimates highlight a need for further research before any strong conclusions are drawn. We also find that under routine service costing, adding a grant linkage component to parenting support is likely to be more cost-effective than parenting support alone. Across the three interventions, the most cost-effective option is likely to be the community grant outreach intervention. We find that if trial-based costing is used for interventions, parenting support and parenting support plus grant linkage are unlikely to be cost-effective at either a population-average or high prevalence of violence, and community grant outreach may only be cost-effective when the prevalence of violence is high among beneficiary households.

The findings from our analysis provide early evidence around the cost-effectiveness of a community outreach intervention to support households' access to social grants if they are eligible but not receiving them. The cost of the proposed grant outreach intervention per adolescent is equivalent to the value of three child support grants, and falls on the lower end of costs for home-visiting interventions in high-income settings [45]. Our findings around the cost-effectiveness of parenting support implemented using routine service costing match a previous economic evaluation of the original PLH Teen intervention in South Africa [11]. In that study, under routine service costing, the cost per DALY averted of parenting support was estimated to be USD 2650 [11]. Our assumption that the modelled routine service intervention is just as effective as the trial-based intervention, is supported by evidence from a pre- and post-test evaluation of PLH Teens implemented with routine service costing [46]. The increasing roll-out and scale-up of parenting support programmes across sub-Saharan Africa will present further opportunities to validate this [47]. The previous economic evaluation of PLH Teen found that parenting support implemented using trial-based costing may be more cost-effective than we estimated in our model [11]. Our study is likely to have underestimated the effects of parenting support by only modelling one impact pathway, caregiver supervision. Other evidence-based impact pathways include improved caregiver mental health, caregiver alcohol/drug avoidance, and improved economic welfare [48]. We also modelled a more conservative prevalence of adolescent violence among families than the previous economic evaluation [11].

Evidence of the enhanced effectiveness of combining multiple interventions with a 'plus' approach has grown substantially in recent years. However, there have been few attempts to evaluate the cost-effectiveness of these interventions [49, 50]. Our findings of the incremental cost-effectiveness of combining an additional grant linkage session with parenting support

further validates the added value of integrated services for reducing violence against adolescents. Building on this, future research should consider other possible approaches for combining parenting support with an economic strengthening intervention [51–53].

Our modelling study enabled us to consider important questions around the design of three interventions for reducing violence against adolescents. These included how best to build on existing social policies such as South Africa's CSG, the resources needed for implementing interventions at scale, and approaches for combining two interventions to simultaneously promote food security and caregiver supervision [3]. Study strengths included its significantly lower cost compared to running a randomised evaluation of the three evaluated interventions, and also our use of a widely applicable methodology that could be used to generate similar evidence for other interventions and/or settings. Our study also had limitations. While many of our model parameters were drawn from high-quality randomised studies, estimates of the association between protective factors and violence outcomes were based on observational research and may be affected by sources of bias associated with this type of research design, including unmeasured confounding [5]. Our focus on the impact of interventions via food security and caregiver supervision alone did not consider the full complement of protective factors via which selected interventions could act on adolescent violence victimisation. There is evidence that parenting support is also likely to act on adolescent violence via improving caregiver mental health, caregiver alcohol/drug avoidance, and improved economic welfare [48, 54, 55]. Gaps in the academic literature also meant that we had to make assumptions about some of the parameters in our model, for example, the post-intervention decay in the effectiveness of cash grants and parenting support programmes. Several methodological choices also mean that our estimates of cost-effectiveness are likely to be conservative. These include our focus on DALYs averted for measuring cost-effectiveness, which while allowing us to generate a single estimate of cost-effectiveness for selected interventions, also meant we could not consider community violence victimisation in our model. Due to a lack of evidence in the wider literature, we were also unable to consider the wider societal benefits of preventing adolescent violence. Such benefits might include reduced health service use, social service use, and court case time. Finally, all three of the interventions considered in this study are likely to have benefits for adolescent development beyond reducing violence victimisation, for example by boosting school enrolment or promoting mental health. Impacts across these additional domains of development should be considered for accurately valuing intervention cost-effectiveness in the future. Doing so may also support cross-sectoral buy-in from multiple government departments, including health, education, and social development [56].

In South Africa, 40% of young people are estimated to experience some form of emotional, physical, or sexual abuse in their lives [57]. Reviewing and analysing the literature on violence against adolescents in South Africa, our study provides novel and policy-relevant evidence for much-needed action to reduce these forms of abuse against adolescents. Our findings suggest that all three of our selected interventions may be cost-effective so long as they are implemented using routine service costing and reach adolescents with the highest risk of violence victimisation. The study also provides a comprehensive summary of evidence gaps that should be investigated as a priority for further informing the scale-up of interventions in this field. These include limited estimates of the prevalence of adolescent violence victimisation among the most vulnerable groups, the effects of cash transfers on adolescent violence victimisation [58, 59], the mechanisms via which cash transfers act on this outcome, and the wider benefits of reducing violence victimisation to communities and society. Future attempts to quantify the cost-effectiveness of social interventions should also account for their effects on other areas of adolescent development, including education and health.

## Conclusion

Investments in grant outreach, and parenting support are most likely to be cost-effective for reducing adolescent violence victimisation if they are provided using routine service costing and they reach adolescents at high risk of violence victimisation. We also find that under routine service costing, adding a grant linkage component to parenting support is likely to be more cost-effective than parenting support alone. Further research to account for all of the pathways via which interventions act to reduce adolescent violence victimisation is necessary to strengthen our confidence in the findings.

## Supporting information

**S1 Checklist. CHEERS checklist.**
(DOCX)

**S1 Fig.** Cost-effectiveness plane scatter plots of costs per adolescent over effectiveness (in DALYs averted) for four scenarios with the SA GDP per capita as the willingness-to-pay threshold at: A) Routine service costing and population-average prevalence of violence, B) Routine service costing and high prevalence of violence, C) Trial-based costing and population-average prevalence of violence, and D) Trial-based costing and population-high prevalence of violence.
(DOCX)

**S2 Fig. Cost-effectiveness plane scatter plots of costs per adolescent over effectiveness (in DALYs averted) for four scenarios using estimates for the relationship between household income and food security drawn from secondary analysis of the South Africa's 2018 GHS.** The SA-specific threshold per capita is used as the willingness-to-pay threshold at: A) Routine service costing and population-average prevalence of violence, B) Routine service costing and high prevalence of violence, C) Trial-based costing and population-average prevalence of violence, and D) Trial-based costing and population-high prevalence of violence.
(DOCX)

**S1 Table. Description of hypothesised interventions, their duration, and staffing structure.**
(DOCX)

**S2 Table. Summary of data sources used to estimate the effectiveness and cost of grant outreach, parenting support, and parenting support plus grant linkage.**
(DOCX)

**S3 Table. Estimated DALY per case of physical, emotional, and sexual abuse in South Africa.**
(DOCX)

**S4 Table. Detailed breakdown of the cost of grant outreach, parenting support, and parenting support plus grant linkage in United States dollars.**
(DOCX)

**S1 Text. Synthesis of the effect of cash grants on household food insecurity in sub-Saharan Africa.**
(DOCX)

**S2 Text. Analysis of the relationship between household income and food security in South Africa using data from the 2018 General Household Survey.**
(DOCX)

## Acknowledgments

We thank Dr Jamie Lachman for providing us with costing data from past implementations of parenting programmes. We thank Dr Yulia Shenderovich for providing data from the parenting for Lifelog Health (PLH Teen) trial. We also thank Gloria Khoza (UNICEF South Africa) and Mpume Danisa (Clowns without Borders South Africa), whose experience in scaling up parenting programmes in South Africa provided valuable insight into the intervention scenarios during the early phases of this work.

## Author Contributions

**Conceptualization:** William E. Rudgard, Sopuruchukwu Obiesie, Chris Desmond, Marisa Casale, Lucie Cluver.

**Data curation:** William E. Rudgard, Sopuruchukwu Obiesie.

**Formal analysis:** William E. Rudgard, Sopuruchukwu Obiesie.

**Funding acquisition:** Chris Desmond, Lucie Cluver.

**Investigation:** William E. Rudgard, Sopuruchukwu Obiesie.

**Methodology:** William E. Rudgard, Sopuruchukwu Obiesie, Chris Desmond, Lucie Cluver.

**Project administration:** William E. Rudgard.

**Supervision:** William E. Rudgard, Chris Desmond, Marisa Casale, Lucie Cluver.

**Validation:** William E. Rudgard, Chris Desmond, Marisa Casale.

**Visualization:** William E. Rudgard, Sopuruchukwu Obiesie.

**Writing – original draft:** William E. Rudgard, Sopuruchukwu Obiesie.

**Writing – review & editing:** William E. Rudgard, Sopuruchukwu Obiesie, Chris Desmond, Marisa Casale, Lucie Cluver.

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
