## [Decision Letter · Decision Letter 0]

6 Mar 2023

PGPH-D-23-00203

Assessing the cost-effectiveness of economic strengthening and parenting support for preventing violence against adolescents in South Africa: An economic modelling study using non-randomised data.

Dear Dr. Rudgard,

Thank you for submitting your manuscript to PLOS Global Public Health. After careful consideration, we feel that it has merit but does not fully meet PLOS Global Public Health’s publication criteria as it currently stands. Therefore, we invite you to submit a revised version of the manuscript that addresses the points raised during the review process, ensuring that all modeling decisions are carefully justified and fully described.

We look forward to receivi

ng your revised manuscript.

Kind regards,

Hannah Hogan Leslie, PhD

Academic Editor

Journal Requirements:

1. Please insert an Ethics Statement at the beginning of your Methods section, under a subheading 'Ethics Statement'. It must include:

1) The name(s) of the Institutional Review Board(s) or Ethics Committee(s)

2) The approval number(s), or a statement that approval was granted by the named board(s) 

3) (for human participants/donors) - A statement that formal consent was obtained (must state whether verbal/written) OR the reason consent was not obtained (e.g. anonymity). NOTE: If child participants, the statement must declare that formal consent was obtained from the parent/guardian.

Additional Editor Comments (if provided):

Reviewers' comments:

Reviewer's Responses to Questions

**Comments to the Author**

1. Does this manuscript meet PLOS Global Public Health’s publication criteria? Is the manuscript technically sound, and do the data support the conclusions? The manuscript must describe methodologically and ethically rigorous research with conclusions that are appropriately drawn based on the data presented.

Reviewer #1: Partly

Reviewer #2: No

2. Has the statistical analysis been performed appropriately and rigorously?

Reviewer #1: N/A

Reviewer #2: No

3. Have the authors made all data underlying the findings in their manuscript fully available (please refer to the Data Availability Statement at the start of the manuscript PDF file)?

Reviewer #1: No

Reviewer #2: Yes

4. Is the manuscript presented in an intelligible fashion and written in standard English?

Reviewer #1: Yes

Reviewer #2: Yes

5. Review Comments to the Author

Reviewer #1: Reviewer #: Article Number: PGPH-D-23-00203.

Article Title: Assessing the cost-effectiveness of economic strengthening and parenting support for preventing violence against adolescents in South Africa: An economic modelling study using non-randomised data.

In this article, the authors used a decision-analytic model to assess the cost-effectiveness of three interventions (linking eligible households to anti-poverty cash grants, group-based parenting support and a combination of the two interventions - group-based parenting support ‘plus’ linkage of eligible households to anti-poverty cash grants) to prevent violence against children in low- and middle-income countries. Using modelling to assess the cost-effectiveness of interventions to prevent violence against children is of great importance as experimental studies, though recommended are costly and complex especially if more than one intervention is being studied. The evaluations of costs and effects presented by the authors can assist public health authorities.

The model that authors use is suitable for the issue addressed, however the authors did not give full details of the model which makes it difficult to understand how the results were obtained. They mention model parameters in the document, but the parameters are not listed in one table which makes it difficult for the reader.

My main considerations are as follows:

Title: The title indicates that the study is on cost-effectiveness of economic strengthening and parenting support for preventing violence against adolescents in South Africa. However, the study is based on data from Mpumalanga province one of the nine provinces in South Africa. Though the authors indicate that Mpumalanga province lie between (in terms of its economic characteristics) the other two provinces Eastern and Western Cape, I feel that the population and economic characteristics in the Mpumalanga province is not a representative of the population in South Africa as a country hence results obtained may not be generalizable to South Africa. I suggest editing the title to reflect that this study is for Mpumalanga province in South Africa.

Introduction: The introduction is quite long (I suggest editing this section). Lines 52-62 can be in the discussion section.

Line 50 to 51 – I suggest deleting this sentence in the introduction. Rather have it in the discussion section as a limitation to this study.

Method section:

1. The use the word intervention (Lines 112, 122 and 132) is confusing. I suggest using scenario instead of intervention. Each scenario will focus on an intervention or a combination of interventions.

2. Fig 1. Graphical representation of probability tree model for estimating the effectiveness of interventions on adolescent violence outcomes – more information is needed in this figure. The structure of a tree is not evident. No probabilities are shown in the tree diagram. To enhance understanding of the methods section all necessary information must appear in the tree diagram or must be well explained in the main text. As it stands now it is not easy to replicate this study. This section is the backbone of this research hence it is necessary to show all the information. Alternatively share references where detailed information about the probability tree model can be found.

3. It might be necessary to justify the discount rate used in this analysis.

Results section:

1. Line 241 – authors mention that the estimates are for 10 years of intervention. However, there is no period stated for results presented before this line. Does this mean that grant outreach intervention was done over 10 years?

2. Line 248 – authors say that grant outreach intervention would be expected to reduce the probability of sexual abuse by -0.54ppts (-0.86; -0.16). If the word reduce is used, the negative sign must be deleted. Also edit the sentences after line 248 (all instances where the word reduce/decrease is used).

3. There are values quoted in the text which do not appear in the main article tables (see lines 293 to 294). I suggest adding all values discussed in the manuscript in tables appearing in the main text.

Discussion section:

1. The low intervention scenario is only mentioned in the discussion section. No mention of such scenario is in the methods section. I suggest adding information on all scenarios in the methods section.

Minor Essential Revisions

1. Page 3 Affiliations: The 4th affiliation is not linked to any author. Please edit

2. Line 25 Abstract: delete “only”

3. Line 27 to 28 Abstract: Last sentence is confusing – cost-effectiveness analysis show that combining the two interventions is second to linking households to grants cost-effective. How does adding a grant linkage component to parenting support enhance cost-effectiveness.

4. Line 95 Study setting: Add “South Africa” after “Mpumalanga province”

5. Line 164 Fig 1: I suggest making the + sign a superscript.

6. Line 272 Parenting support: It should be USD36 (as in Table 3) instead of USD37.

7. Line273-274 Parenting support: It’s not clear where the following values are coming from “(USD 275,700 and 1,640,000 annually)”.

8. Line 280: Value (2 617 000) is different from what is in Table 3.

Clear labelling for all figures is required. Generally, grammar is good despite minor grammatical errors picked in the document. Reading through the document after final revision is recommended.

Supplementary file

Authors are commended for submitting a comprehensive supplementary file. I have a few comments to make

1. CHEERS list:

Choice of model – this point is not fully addressed in the manuscript. See Method section comments above.

Study parameters – not all parameters are in Table 1.

2. Page 11 Use ‘INSPIRE’ instead of ‘Inspire’

3. Since the study is based on Mpumalanga province. Is it necessary to list provincial aOR’s for all provinces?

4. S8: Which one is the lean and ideal scenario? In the main text authors use trial and routine service delivery. The Low effect scenario is not described in the manuscript.

Reviewer #2: The authors conducted a cost-effectiveness analysis of the individual and joint effects of two interventions (outreach+linkage to cash transfer and parenting support) on violence experiences among South African adolescents. I enjoyed reading the paper and feel it makes interesting contributions to support violence prevention policy and prioritization. However, I identified several weaknesses, some fairly major, and points of clarification that should be addressed, as outlined below:

1. The violence domains the authors explore in this paper could be better and more explicitly defined in the introduction/methods. The introduction mentions adolescents ‘affected by violence,’ but the methods and discussion section more fluidly mixes a bunch of violence outcomes: victimization (e.g. physical and sexual abuse), perpetration (e.g. youth lawbreaking), and an item that is unclear in its mapping to violence (transactional sex). It becomes clearer later on that the authors are only looking at emotional, physical, and sexual abuse, but this seems to be a practical restriction rather than one made based on domains/modes of intervention efficacy. Suggest more clearly defining area of outcome focus throughout. Specifically the mention of perpetration outcomes and transactional sex in the methods and discussion is confusing.

2. Importantly, Interventions 1 and 3 (grant outreach) are only able to assist households that do not already access the Child Support Grant, but this is only a small percentage of the target population. It also seems like the models are parameterized with intervention effects of the cash transfer. So there is a mixture of interventions and parameterization here that is confusing. I think the CSG is actually the intervention you want to model, in target populations with varying levels of pre-existing uptake. It seems that this is also the cost that is most relevant: the cost of the actual CSG program, not just the cost to link people to the program.

3. Adolescents age 18 and 19 are not eligible for the Child Support Grant (coverage only goes until age 18), but the target population includes adolescents age 11-19?

4. The Child Support Grant is modeled as acting on violence outcomes via food security, which seems rather narrow. The CSG could also plausibly work through, for example, education, mental health, and stress pathways, among others. Similar point is probably true for parenting interventions. Do you allow any of these interventions to work through other pathways besides the limited protective factors specified (food security and positive parenting)? It seems a major underestimate to restrict these holistic interventions to only their effects through narrow intermediate pathways and not their broader set of direct and indirect effects.

5. On page 7: “For parenting support plus grant linkage, we used our pooled estimate for the effect of cash transfers on food security and our effect estimate for caregiver supervision from the PLH Teen evaluation.” It’s unclear to me whether this pooled estimate incorporates any potential interaction between the two interventions or just adds their effect together?

6. The 10 year implementation timeline strikes me as a bit odd – people in your target population would age in and out of adolescence and out of CSG eligibility over that timeframe.

7. How is it possible that the grant outreach alone prevents more cases of violence than the grant outreach + parenting support (Table 3)?

8. The conversion of ‘Averted cases of violence’ to DALYs is not presented clearly in the results section.

9. The conclusions in the discussion “Our model indicates that implementing each of our three selected interventions using a routine service delivery model is likely to be cost-effective for reducing emotional, physical, and sexual abuse against adolescents” and the abstract “Findings indicate that investments in community grant outreach, and parenting support interventions are likely to be cost-effective for preventing adolescent violence” don’t seem to align with the findings as reported in the results section. The results seemed to indicate that, by and large, the interventions were not cost-effective (below the WTP threshold for all except under assumptions of high violence)? So I’m not clear on how these conclusions were arrived at.

Minor:

10. The language around ‘incurring’ DALYs in the intro seems off. My understanding is that they are a measure of years of healthy life lost – suggest updating the language to better align with its definition.

11. Small typos and missing words throughout – needs copyediting.

6. PLOS authors have the option to publish the peer review history of their article (what does this mean?). If published, this will include your full peer review and any attached files.

**Do you want your identity to be public for this peer review?** For information about this choice, including consent withdrawal, please see our Privacy Policy.

Reviewer #1: No

Reviewer #2: No

---

## [Decision Letter · Decision Letter 1]

10 Jul 2023

Assessing the cost-effectiveness of economic strengthening and parenting support for preventing violence against adolescents in Mpumalanga Province, South Africa: An economic modelling study using non-randomised data.

PGPH-D-23-00203R1

Dear Mr. Rudgard,

We are pleased to inform you that your manuscript 'Assessing the cost-effectiveness of economic strengthening and parenting support for preventing violence against adolescents in Mpumalanga Province, South Africa: An economic modelling study using non-randomised data.' has been provisionally accepted for publication in PLOS Global Public Health. Please note two minor requests for clarification below as you prepare for publication.

Best regards,

Hannah Hogan Leslie, PhD

Academic Editor

Please address the minor points raised on re-review in the process of preparing the manuscript for publication.

Reviewer Comments (if any, and for reference):

Reviewer's Responses to Questions

**Comments to the Author**

1. If the authors have adequately addressed your comments raised in a previous round of review and you feel that this manuscript is now acceptable for publication, you may indicate that here to bypass the “Comments to the Author” section, enter your conflict of interest statement in the “Confidential to Editor” section, and submit your "Accept" recommendation.

Reviewer #1: All comments have been addressed

Reviewer #2: All comments have been addressed

2. Does this manuscript meet PLOS Global Public Health’s publication criteria? Is the manuscript technically sound, and do the data support the conclusions? The manuscript must describe methodologically and ethically rigorous research with conclusions that are appropriately drawn based on the data presented.

Reviewer #1: Yes

Reviewer #2: Yes

3. Has the statistical analysis been performed appropriately and rigorously?

Reviewer #1: N/A

Reviewer #2: Yes

4. Have the authors made all data underlying the findings in their manuscript fully available (please refer to the Data Availability Statement at the start of the manuscript PDF file)?

Reviewer #1: Yes

Reviewer #2: (No Response)

5. Is the manuscript presented in an intelligible fashion and written in standard English?

Reviewer #1: Yes

Reviewer #2: Yes

6. Review Comments to the Author

Reviewer #1: The authors are commended for addressing all the comments and making detailed revisions to the manuscript. The clarity and organisation of the manuscript has significantly improved. The introduction provides a clearer background and rationale for the study. The Methods section has been revised to include more details on the model and the probability tree. Tables and figures in the manuscript and supplementary file are clear and appropriately labelled, aiding in the understanding of the analysis. The authors have added information on the limitations of the study. I recommend acceptance of manuscript pending two minor comments/revisions outlined below:

1. Line 86 to 88: “Of the four violence outcomes investigated in the non-randomised analysis in South Africa, this included emotional, physical, and sexual abuse, but excluded community violence victimization (5).” This sentence needs revision. I think there is something missing after the word “this” maybe “…this study included….”

2. Table 1 – Authors have 1,105,791 in the value column against Eligible children receiving a CSG but 18% of the Total population is 814 140 – (first two rows in the table which should be the total Adolescent population in Mpumalanga). How is it then possible to have 1,105,791 eligible children receiving a CSG. May the authors have a second look on these data. I might be interpreting the data incorrectly.

Other than the minor comments above I recommend acceptance of manuscript

Reviewer #2: This response was a pleasure to read. The authors have thoroughly responded to all prior comments and I have no further concerns.

7. PLOS authors have the option to publish the peer review history of their article (what does this mean?). If published, this will include your full peer review and any attached files.

**Do you want your identity to be public for this peer review?** For information about this choice, including consent withdrawal, please see our Privacy Policy.

Reviewer #1: No

Reviewer #2: **Yes: **Molly Rosenberg
